# Surveillance for Patients with Oral Squamous Cell Carcinoma after Complete Surgical Resection as Primary Treatment: A Single-Center Retrospective Cohort Study

**DOI:** 10.3390/cancers13225843

**Published:** 2021-11-21

**Authors:** Chonji Fukumoto, Ryo Oshima, Yuta Sawatani, Ryo Shiraishi, Toshiki Hyodo, Ryouta Kamimura, Tomonori Hasegawa, Yuske Komiyama, Sayaka Izumi, Atsushi Fujita, Takahiro Wakui, Hitoshi Kawamata

**Affiliations:** 1Department of Oral and Maxillofacial Surgery, Dokkyo Medical University School of Medicine, Mibu 321-0293, Tochigi, Japan; chonji-f@dokkyomed.ac.jp (C.F.); r-oshima@dokkyomed.ac.jp (R.O.); sawayu@dokkyomed.ac.jp (Y.S.); ryo-s@dokkyomed.ac.jp (R.S.); hyodo14@dokkyomed.ac.jp (T.H.); kmry28@dokkyomed.ac.jp (R.K.); hase-t@dokkyomed.ac.jp (T.H.); y-komi@dokkyomed.ac.jp (Y.K.); saya@dokkyomed.ac.jp (S.I.); fujita-a@dokkyomed.ac.jp (A.F.); t-wakui@dokkyomed.ac.jp (T.W.); 2Department of Oral and Maxillofacial Surgery, Kamma Memorial Hospital, Nasushiobara 325-0046, Tochigi, Japan; 3Section of Dentistry and Oral and Maxillofacial Surgery, Kami-Tsuga General Hospital, Kanuma 322-8550, Tochigi, Japan

**Keywords:** oral squamous cell carcinoma, surveillance, recurrence, metastasis, positron emission tomography/computed tomography (FDG-PET/CT), contrast-enhanced computed tomography (CECT)

## Abstract

**Simple Summary:**

Surveillance methods for oral squamous cell carcinoma (OSCC) patients may be chosen by considering the risk of recurrence. We performed a retrospective cohort study in 324 OSCC patients after complete surgical resection as the primary treatment from 2007 to 2020 at our hospital. Regarding the time to occurrence of postsurgical events, we found that half of cases of local recurrence, cervical lymph node, and distant metastasis occurred within 200 days, and 75% of all these events occurred within 400 days. However, the mean time for second primary cancer was 1589 days. The postsurgical events were detected by imaging examinations earlier than they were by visual examination and palpation. It is desirable to perform FDG-PET/CT within 3–6 months and at 1 year after surgery and to consider CECT as an option in between FDG-PET/CT, while continuing history and physical examinations for about 5 years based on an individual risk assessment.

**Abstract:**

Background: The surveillance methods oral squamous cell carcinoma (OSCC) patients may be chosen by considering the risk for recurrence, and it is important to establish appropriate methods during the period in which latent/dormant cancer cells become more apparent. To investigate the appropriate surveillance of patients with OSCC based on the individual risk for recurrence and/or metastasis, we performed a retrospective cohort study after the complete surgical resection of OSCC as the primary treatment. Methods: The study was performed in 324 patients with OSCC who had been primarily treated with surgery from 2007 to 2020 at our hospital. We investigated the period, timing, and methods (visual examination, palpation and imaging using FDG-PET/CT or CECT) for surveillance in each case that comprised postsurgical treatment. Results: Regarding the time to occurrence of postsurgical events, we found that half of cases of local recurrence, cervical lymph node metastasis, and distant metastasis occurred within 200 days, and 75% of all of these events occurred within 400 days. However, the mean time for second primary cancer was 1589 days. The postsurgical events were detected earlier by imaging examinations than they were by visual examination and palpation. Conclusions: For the surveillance of patients with OSCC after primary surgery, it is desirable to perform FDG-PET/CT within 3–6 months and at 1 year after surgery and to consider CECT as an option in between FDG-PET/CT, while continuing history and physical examinations for about 5 years based on individual risk assessment.

## 1. Introduction

Oral squamous cell carcinoma (OSCC) is mainly treated with surgery in combination with radiotherapy, chemotherapy (including molecular-targeted drugs), and immune checkpoint inhibitors. Although these treatments have improved the prognosis of patients with OSCC over the years, local recurrence, cervical lymph node metastasis, and distant metastasis may be observed after initial complete resection. The guidelines for head and neck cancer by the National Comprehensive Cancer Network (NCCN) indicate extranodal extension, positive margins, close margins, pT3 or pT4 primary, pN2 or pN3 nodal disease, nodal disease in levels IV or V, and perineural, vascular, and lymphatic invasion as adverse features (AFs) for a poor postsurgical outcome [1]. We extracted extranodal extension, positive margins, close margins, pN2 or pN3 nodal disease, and nodal disease in levels IV or V among the AFs and defined them with the addition of the Yamamoto–Kohama (YK) mode of invasion [2,3], with YK-4C and YK-4D as the major risk factors for recurrence or metastasis [4]. We have used these risk factors since 2014, and we have found that patients with these factors have significantly poorer outcomes and that these outcomes are significantly improved using combination therapy with cetuximab and paclitaxel [4,5]. This postsurgical therapy with cetuximab is a preemptive therapy for dormant tumors that cannot be visualized, meaning that it might not deviate from the NCCN guidelines. On the other hand, although the NCCN guidelines state that “postsurgical treatment should be considered” for pT4a and pN1, we have not applied postsurgical treatment for patients with pT4a or pN1 alone. This is because we think that pT4a (local tumor size and progression) are not prognostic factors if complete surgical resection is performed with adequate surgical margins. Therefore, we do not include locally advanced cancer in the major risk category. Furthermore, we think that the biological aggressiveness of the tumor and prognosis between >pN2 patients and pN1 patients are quite different. Therefore, we did not include pN1 patients in the major risk category.

The period and the surveillance methods patients for with OSCC after primary treatment, including surgery, are important to extract clinical problems. OSCC is most likely to recur in the first 2 years after treatment, but patients should be observed for at least 5 years posttreatment [1,6]. The detection of local recurrence, cervical lymph node metastasis, distant metastasis, and second primary cancers as early as possible post-surgery is important for selecting the salvage therapy and for improving the outcome. In OSCC cases with postsurgical events, we hypothesized that latent/dormant cancer cells that were undetectable before or during the initial treatment might grow and become visible tumors after the primary treatment [4,5].

History and physical examinations are important for the detection of these events, but the occurrence of the events in the deep space in the body cannot be detected through visual examination or palpation through the surface. Therefore, adequate imaging examinations are necessary to detect events in the deep space in the body. ^18^F-fluorodeoxyglucose (FDG)-positron emission tomography (PET)/computed tomography (CT), contrast-enhanced CT (CECT), and magnetic resonance imaging (MRI) are useful for viewing the primary OSCC lesions and metastases [1,6,7,8,9]. However, some imaging has a risk for radiation exposure, an inability to detect certain events, and medical costs. Therefore, the surveillance methods may be chosen by means of an evaluation of the risk of recurrence, and it is important to establish appropriate methods in the period in which latent/dormant cancer cells become more apparent. However, in the NCCN guidelines, surveillance is only described as a follow-up recommendation, and it is also mentioned that standardized multicenter trials are needed to examine the value of routine imaging in clinically asymptomatic patients [1]. Surveillance for cervical lymph node metastasis after systemic chemotherapy/radiotherapy (RT) or RT only is presented as a flow chart in the NCCN guidelines, but it only shows the recommended surveillance period, the frequency, and the advantages of each imaging examination after the surgical treatment of local advanced cancer [1]. To clarify the appropriate period, timing, and methods for the surveillance of patients with OSCC based on individual risk for recurrence and/or metastasis, we performed a retrospective cohort study after the complete surgical resection of OSCC as the primary treatment. 

## 2. Patients and Methods

### 2.1. Data Sources

A retrospective cohort study was performed in OSCC patients treated at the Department of Oral and Maxillofacial Surgery, Dokkyo Medical University Hospital from 2007 to 2020. We obtained the data from electronic medical records. 

### 2.2. Patients

We collected the data for 324 OSCC patients who had undergone surgery as primary treatment during the study period. We confirmed the complete surgical resection of the visible tumor with a safety margin in all cases. In a case where a patient had a resected tumor on the surgical margin or close to the surgical margin (within 5 mm), we performed additional resections on those patients within 1 week after the determination of the status of the surgical margin. The UICC TNM Classification of Malignant Tumours, 8th ed. [10] was used to determine cancer status. Extranodal extension (ENE), positive margins, close margins, and pN2 or pN3 nodal disease in the AFs of the NCCN Guidelines for Head and Neck Cancers [1] with the addition of the YK-4C or YK-4D stage [2,3] were defined as major risk factors. If at least one of these factors was met, then we categorized the patient as being at major risk for a poor outcome.

Since the study was performed over the course of 14 years, treatment strategies varied over time, but general clinical factors were examined for the patients who were at major risk for a poor outcome and in cases where postsurgical treatment was considered. In the early part of the study period, this treatment was mainly platinum-based chemotherapy alone or combined with RT, as described in the NCCN guidelines, but after the introduction of cetuximab, postsurgical treatment including cetuximab became common as a preemptive therapy for dormant cancer cells [4]. After the completion of postsurgical treatment, the oral administration of S-1 was recommended for all patients as adjuvant therapy at 1 year after surgery [11].

We investigated the period, timing, and methods (visual examination, palpation and imaging using FDG-PET/CT or CECT) of surveillance in each case. As per our surveillance protocols, visual examinations should be performed at least once per month until 1 year after surgery, at least once every 2 months from 1 to 2 years after surgery, and once every 3–6 months from 3 to 5 years after surgery, which is similar to the examination intervals recommended in the NCCN guidelines [1]. As for imaging, CECT should be performed about 1 month after surgery as a baseline, followed by FDG-PET/CT or CECT every 6 months to 1 year thereafter, with FDG-PET/CT performed once a year until 5 years after surgery. The CT range is set to the parietal region over the base of the lung for the screening of both local and cervical lymph nodes and the lung field. FDG-PET/CT was performed with CECT in some cases and with plain CT in others.

Postsurgical events were defined as local recurrence, cervical lymph node metastasis, distal metastasis, and second primary cancer. Times to death and occurrence of postsurgical events were calculated from the date of the operation, regardless of the use of postsurgical treatment.

### 2.3. Statistical Analysis

The 5-year OS and disease-free (DF) rates were evaluated in each clinical cancer stage, major and minor risk groups, and for each postsurgical event using Kaplan–Meier survival analysis. Significant differences were evaluated by the Log rank test. Multivariate analysis of outcomes and postsurgical events was performed using YK-4C and 4D, positive or close margin, metastasis to ≥2 lymph nodes (LNs), and ENE as potential risk factors. Hazard ratios (HRs) for these factors were obtained using univariate and multivariate analyses with a Cox proportional hazard model. Box-and-whisker plots were used to compare the periods between types of postsurgical events, examinations that detected these events, major/minor risk groups, and metastasis to ≥2 LNs (yes/no). A Student t-test was used to compare the means between these groups. Two-tailed *p* values of <0.05 were considered to be significant. IBM SPSS ver. 24.0 (IBM SPSS, Inc., Tokyo, Japan) was used for all of the statistical analyses.

## 3. Results

### 3.1. Characteristics and Treatment of Patients with OSCC

The characteristics and treatment of the 324 patients with OSCC are shown in Table 1. The patients included 190 males (58.6%), and the median age was 69 years. The most frequent primary site was the tongue (*n* = 156, 48.1%), followed by the lower gingiva (*n* = 69, 21.3%). The clinical stages were cT4a (*n* = 114, 35.2%), cN0 (*n* = 190, 58.6%) and cN1 (*n* = 65, 20.1%), and cStage 4a (*n* = 127, 39.2%) and cStage 3 (*n* = 66, 20.4%). Pathological extranodal expansion (*n* = 26, 8.0%), positive or close margins (*n* = 12, 3.7%), metastasis to ≥2 LNs (*n* = 41, 12.7%), and YK-4C or YK-4D (*n* = 32, 9.9%) were present as major risk factors. Treatment included surgery alone (*n* = 254, 78.4%) and a combination of surgery and postsurgical treatment (*n* = 70, 21.6%), mainly with cisplatin (*n* = 22) and cetuximab + paclitaxel (*n* = 15). Adjuvant therapy was used in 129 patients (39.8%). Postsurgical events occurred in 65 patients (20.1%), and 55 patients (17.0%) died from a non-disease-specific cause within 5 years after surgery.

### 3.2. Prognosis after Surgical Treatment

The cumulative 5-year OS and DF rates in each clinical stage are shown in Figure 1. At stages 0, 1, 2, 3, 4A, and 4B, the 5-year OS rates (Figure 1A) were 100%, 81.5%, 88.4%, 85.0%, 66.3%, and 80.0%, respectively (*p* < 0.001); the 5-year DF rates (Figure 1B) were 100%, 98%, 94%, 77.5%, 72.5%, and 66.7%, respectively (*p* < 0.001). Patients in the major risk group had a significantly lower 5-year OS (Figure 2A) (61.9% vs. 85.2%, *p* < 0.001) and 5-year DF rate (Figure 2B) (65.7% vs. 89.9%, *p* < 0.001). The Kaplan–Meier survival curves differed for the OS rate for about 600 days (Figure 2A) and for the DF rate for about 400 days (Figure 2B). After these times, each curve was similar in the major and minor risk groups.

### 3.3. Mortality and Postsurgical Events in Patients with Major Risk Factors

Mortality rates based on major risk factors are shown in Table 2. Mortality did not differ significantly for YK-4C/4D (univariate analysis: HR 1.068, 95% CI 0.426–2.681, *p* = 0.888; multivariate analysis: HR 1.276, 95% CI: 0.504–3.230, *p* = 0.607). Furthermore, mortality did not differ significantly for positive or close margin (univariate analysis: HR 1.161, 95% CI 0.283–4.764, *p* = 0.836; multivariate analysis: HR 1.401, 95% CI 0.339–5.786, *p* = 0.641). Mortality significantly increased based on ENE in univariate analysis (HR 3.285, 95% CI 1.696–6.366, *p* < 0.001) but not in multivariate analysis (HR 1.676, 95% CI 0.769–3.650, *p* = 0.194). However, mortality was significantly increased by metastasis to ≥2 LNs in univariate (HR 4.006, 95% CI 2.278–7.043, *p* < 0.001) and multivariate (HR 3.342, 95% CI 1.720–6.492, *p* < 0.001) analyses. In all of the patients with high YK classification (YK-4C and 4D), surgical treatment (primary resection and neck dissection) might be appropriately performed, and the application of cetuximab-based therapy as a preemptive treatment for this group improved the prognosis.

The occurrence of postsurgical events based on major risk factors are shown in Table 3. The occurrence rate of the event did not differ significantly for YK-4C/4D (univariate analysis: HR 0.853, 95% CI 0.426–2.528, *p* = 0.908; multivariate analysis: HR 1.136, 95% CI 0.403–3.201, *p* = 0.810) or for positive or close margin (univariate analysis: HR 1.992, 95% CI 0.619–6.411, *p* = 0.248; multivariate analysis: HR 2.100, 95% CI 0.647–6.817, *p* = 0.217). In contrast, the occurrence rate of the event was significantly higher based on ENE (univariate analysis: HR 4.397, 95% CI 2.287–8.457, *p* < 0.001; multivariate analysis: HR 2.425, 95% CI 1.101–5.339, *p* = 0.028) and metastasis to ≥2 LNs (univariate analysis: HR 4.175, 95% CI 2.289–7.613, *p* < 0.001; multivariate analysis: HR 2.955, 95% CI 1.437–6.078, *p* = 0.003).

### 3.4. Prognosis after Occurrence of Postsurgical Events

The 5-year cumulative OS rates after postsurgical events (Figure 3) were 29.6% after local recurrence alone, 60.6% after cervical lymph node metastasis alone, 0% after distant metastasis alone, and 0% in patients with two or more of these events (*p* < 0.001).

### 3.5. Comparisons of Periods until Occurrence of Postsurgical Events 

The mean times of occurrence of postsurgical events (Figure 4) were 206.5 days in all cases and 411.9, 169.4, and 304 days in cases with local recurrence, cervical lymph node metastasis, and distant metastasis, respectively. Half of the cases occurred within 200 days for all events (quartile 2/4, Q_2/4_: 155.5, 200.5, 137.5, 189), and 75% occurred within 400 days (Q_3/4_: 287, 2352.5, 277, 394). The mean time of occurrence of second primary cancer was 1589.7 days (Q_2/4_: 1153 days, Q_3/4_: 2196 days), and second primary cancer had a significantly later onset than other postsurgical events (*p* < 0.001).

Methods that detected postsurgical events and times of occurrence are shown in Figure 5. The mean time for the detection of events by FDG-PET/CT and CECT was 345.1 days (Q_2/4_: 190.5 days, Q_3/4_: 359.3 days), whereas that for events detected by visual examination and palpation was 839.3 days (Q_2/4_: 294 days, Q_3/4_: 1139.5 days). Events were detected visually at a significantly later time compared to those detected by FDG-PET/CT and CECT (*p* = 0.001).

Differences in the time of occurrence of postsurgical events between patients in the major and minor risk groups are shown in Figure 6A. The mean time of occurrence of postsurgical events of 396.4 days (Q_2/4_: 213 days, Q_3/4_: 625 days) in the major risk group was significantly earlier than that of 747 days (Q_2/4_: 224 days, Q_3/4_: 851.5 days) in the minor risk group (*p* = 0.001). The time until event occurrence was also examined in cases with and without metastasis to ≥2 cervical LNs (Figure 6B). The mean time of 312.2 days (Q_2/4_: 160 days, Q_3/4_: 379.3 days) in patients with ≥2 metastatic LNs was significantly earlier than that of 730.9 days (Q_2/4_: 252 days, Q_3/4_: 820 days) in patients with <2 metastatic LNs (*p* = 0.05). 

## 4. Discussion

In this study, the importance of surveillance for patients with OSCC after complete surgical resection as the primary treatment and the appropriate period, timing, and methods for surveillance were investigated. The 5-year OS and DF rates during the study period were high compared to previous reports [12,13]. A comparison of our previously established risk factors showed that the outcome was significantly poorer and that the occurrence of postsurgical events was higher in patients with these risk factors. We have already reported that the outcome was significantly improved by cetuximab as postsurgical preemptive therapy in these patients [4]. However, postsurgical events still occurred, and the outcome was poorer in patients with risk factors compared to those without these factors who did not receive postsurgical treatment.

Based on these clinical manifestations, we considered the meaning of surveillance for patients with OSCC after complete surgical resection as follows: First, in case with local recurrence, an additional resection of the recurrent tumor might be difficult because we resected the tumor with an adequate surgical margin (at least 10 mm) at any anatomical site during primary surgery and confirmed the negative surgical margin (for close and positive surgical margins, additional resection was immediately performed). Therefore, a recurrent tumor may be from an anatomically unresectable area or may be due to multiple local micrometastasis. In such cases, the salvage rate was low (29.6%). Chemoradiotherapy and immune checkpoint inhibitors are applied for local recurrence in these cases, and such interventions being applied at the time when the patient condition is good and when the tumor is small will increase the probability of controlling the recurrent tumor. Second, in cases with cervical lymph node metastasis, the probability of salvage is high (60.6%) when discovered early, which indicates the importance of strict surveillance. It is uncertain whether preventive neck dissection with primary surgery or a wait-and-see approach is appropriate for cN0 cases [14,15,16]. This might pose a problem for diagnostic accuracy before treatment and during postsurgical surveillance. In our opinion, positive cervical lymph node metastasis diagnosed before treatment with high accuracy should be treated with neck dissection, and no neck dissection should be induced for negative cases. However, since imaging sensitivity is not 100% (93.6% at our facility [17]), false-negative cases can be treated by salvage neck dissection under strict surveillance. Cervical lymph node metastasis discovered within several months after primary surgery is likely to be due to invisible micrometastasis that were present before surgery and that manifested after surgery due to the perioperatively immunosuppressed state, loss of tumor antigen presentation due to resection of the primary tumor, and loss of anti-angiogenic factors from the primary tumor. The detection of a lymph node metastasis after primary surgery as early as possible by surveillance using FDG-PET/CT and CECT is ideal. Third, for distant metastasis, systemic chemotherapy and immune checkpoint inhibitors are often applied, and if such interventions can be performed when the patient condition is good and when the tumor is small, then the probability of extending survival increases [18,19].

Regarding the time to the development of postsurgical events, we found that a second primary cancer had a significantly later onset compared to other events, indicating that it is necessary to separate the surveillance time axis among these events. Half of the cases of local recurrence, cervical lymph node metastasis, and distant metastasis occurred within 200 days after primary surgery, and 75% of all of these events occurred within 400 days. Based on these findings, surveillance for these events should be the focus during the first postsurgical year, and especially in the first half of that year, whereas the surveillance of a second primary cancer should be continued after 5 years. Since a second primary cancer is a new lesion arising due to the carcinogenic risk of the patient, it may be controllable if it is discovered and resected early, similar to the primary lesion.

There are several guidelines for treating patients with head and neck cancer, but we believe that the NCCN guidelines are the most internationally recognized and utilized guidelines. The NCCN guidelines on the surveillance of surgically treated patients suggest follow-up every 1–3 months after 1 year, every 2–6 months after 2 years, every 4–8 months after 3–5 years post-surgery, and every 12 months after 5 years are recommended for a History and Physical (H&P) examination [1]. In addition, locoregionally advanced disease after treatment is divided into cases of <6 months and ≥6 months to 5 years post-treatment, and the utilization and problems of FDG-PET/CT are presented. Following surgery in patients with locoregionally advanced cancer, short-term post-treatment imaging is recommended for those who show signs of early recurrence or who are at a high risk for early recurrence prior to starting adjuvant postsurgical therapy [1]. We compared the times for detection of postsurgical events with visual examination/palpation and imaging (FDG-PET/CT or CECT) for surveillance. The Q_2/4_ and Q_3/4_ of the time for event detection by FDG-PET/CT and CECT were 190.5 and 359.3 days, respectively, showing that about 50 and 75% of cases were detected within half a year and 1 year, respectively. In contrast, the Q_2/4_ and Q_3/4_ of the time of detection of postsurgical events by visual examination/palpation were 294 and 1139.5 days, respectively. Postsurgical events that could be detected by palpation and visual examination were limited to local recurrences confined to areas exposed to the oral cavity and lymph node lesions that could be palpable from the external surface. It was difficult to detect metastatic lymph nodes behind the sternocleidomastoid muscle or the upper area around the internal jugular vein (level II area) by palpation or visual examination. As for endoscopy and ultrasonography, they were considered to be useful methods for surveillance, and we used them routinely. However, as with visual examination and palpation, only surface and sub-surface areas could be monitored using these devices. On the other hand, FDG-PET/CT was known to be effective in detecting viable tumor cells after primary chemoradiation therapy and in primary tumors in early stage cancers of unknown primary. Here, we demonstrated that FDG-PET/CT was also useful for detecting metastatic disease in deep tissues and distant organs in patients after primary surgery. These findings suggest that it is desirable to perform FDG-PET/CT or CECT in the 3- to 6-month period after surgery and to continue visual examination/palpation for more than 5 years.

Our analysis of the risk factors for a poor outcome showed a significant association of metastasis to ≥2 cervical LNs. Thus, surveillance should be carefully performed within 1 year after surgery, especially in the first 6 months, in patients with ≥2 cervical LN metastases. However, Q_2/4_ was also within 9 months in patients without risk factors or without ≥2 cervical LN metastases, which indicates that the frequency of imaging surveillance should not be decreased in these patients.

Although there are several methods for establishing and proving appropriate surveillance methods, we believe that the accumulation of high-quality retrospective and prospective observational studies is important. The results of this study support the recommendations for surveillance in the NCCN guidelines: It is desirable to perform FDG-PET/CT within 3–6 months after surgery and to define the interval of H&P. FDG-PET/CT is a sensitive imaging modality, with 12-month PET revealing recurrent or second primary cancers in about 10% of treated patients and 24-month FDG-PET/CT showing these findings in about 5% of treated cases [1,7]. At 6 months after surgery and thereafter, Ho et al. found no significant difference in the 3-year DF survival rate in patients with OSCC undergoing imaging surveillance or clinical surveillance only (41% vs. 46%, *p* = 0.91) [1,8]. In addition, the NCCN guidelines suggest that there may be little proven benefit in further imaging if the initial 3-month FDG- PET/CT scan is negative. If FDG-PET/CT at 3 months is negative post-surgery, then there are no data to support a substantial benefit for further imaging in an asymptomatic patient [1]. However, event detection by FDG-PET/CT or CECT was also noted even at 2 years after surgery in this study, although the incidence was low, with a Q_3/4_ for event detection of 359.3 days and a maximum excluding outlier of 550 days. Thus, the use of imaging after the period with a high incidence of postsurgical events requires further study.

## 5. Conclusions

For the surveillance of patients with OSCC after primary surgery, it is desirable to perform FDG-PET/CT within 3–6 months and at 1 year after surgery and to consider CECT as an option in between FDG-PET/CT. Furthermore, it is also advisable to adjust the frequency of imaging after 1 year while continuing H&P examinations for about 5 years based on an individual risk assessment.

## Figures and Tables

**Figure 1 cancers-13-05843-f001:**
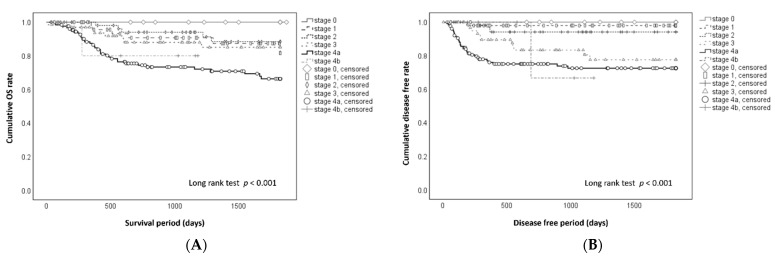
(**A**). Cumulative overall survival (OS) rate in patients with oral squamous cell carcinoma. The 5-year OS rates were 100% at stage 0, 81.5% at stage 1, 88.4% at stage 2, 85.0% at stage 3, 66.3% at stage 4A, and 80.0% at stage 4B (*p* < 0.001). (**B**). Cumulative disease-free (DF) rate in patients with oral squamous cell carcinoma. The 5-year DF rates were 100% at stage 0, 98% at stage 1, 94.1% at stage 2, 77% at stage 3, 72.5% at stage 4A, and 66.7% at stage 4B (*p* < 0.001).

**Figure 2 cancers-13-05843-f002:**
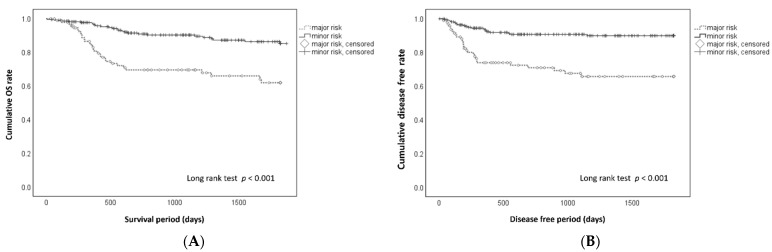
(**A**). Cumulative overall survival (OS) rate in patients with oral squamous cell carcinoma in the major and minor risk groups. The 5-year OS rates were 61.9% in the major risk group and 85.2% in the minor risk group, with significantly lower OS in the major risk group (*p* < 0.001). (**B**). Cumulative disease-free (DF) rate in patients with oral squamous cell carcinoma in the major and minor risk groups. The 5-year DF rates were 65.7% in the major risk group and 89.9% in the minor risk group, with significantly lower DF in the major risk group (*p* < 0.001).

**Figure 3 cancers-13-05843-f003:**
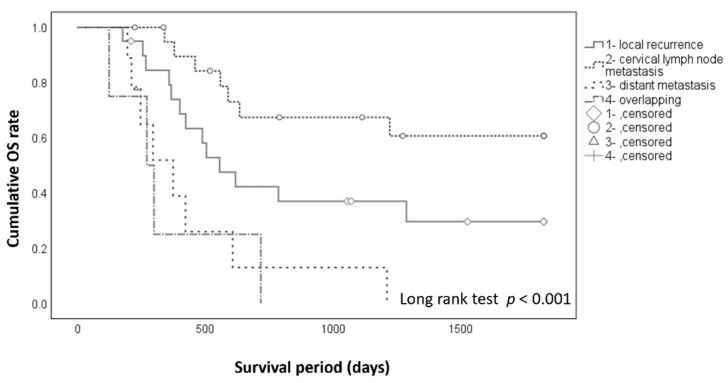
Cumulative overall survival (OS) after postsurgical events. The OS rate after occurrence of events was 29.6% for local recurrence alone, 60.6% for cervical lymph node metastasis alone, 0% for distant metastasis alone, and 0% in patients with two or more of these events (*p* < 0.001).

**Figure 4 cancers-13-05843-f004:**
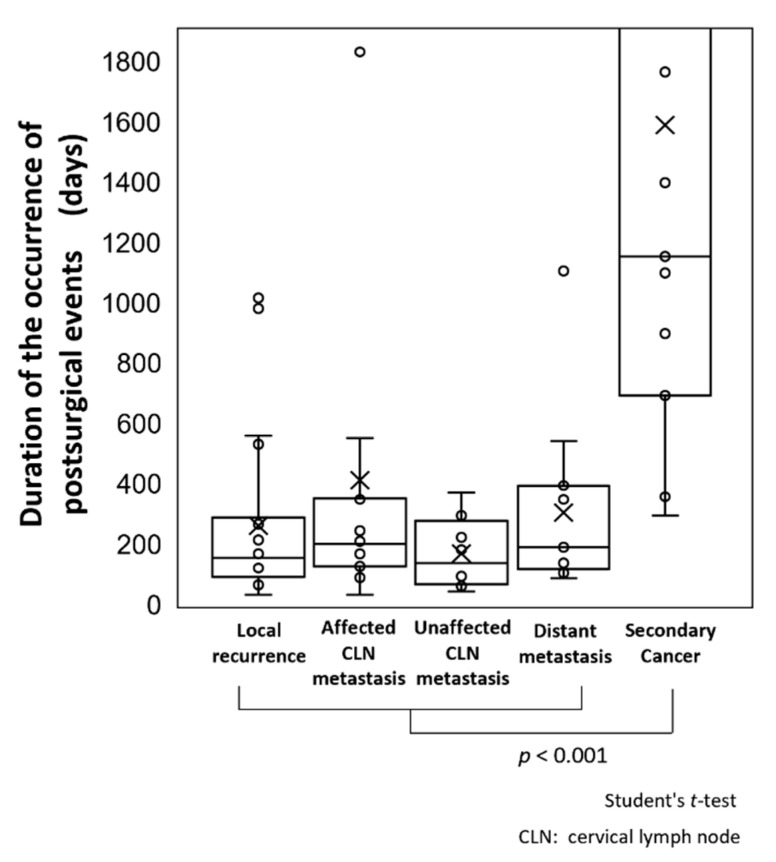
Comparisons of periods until occurrence of postsurgical events. The mean times for occurrence of all postsurgical events and local recurrence, cervical lymph node metastasis, and distant metastasis were 206.5, 411.9, 169.4, and 304 days, respectively. Half of all of the events occurred within 200 days (Q_2/4_ 155.5, 200.5, 137.5, 189), and 75% occurred within 400 days (Q_3/4_ 287, 2352.5, 277, 394). The mean time for the occurrence of second primary cancer was 1589.7 days (Q_2/4_ 1153 days, Q_3/4_ 2196 days), indicating significantly later onset compared with other events (*p* < 0.001). In the figure, “×“ indicates the mean value and “○“ indicates the event occurrence per patient.

**Figure 5 cancers-13-05843-f005:**
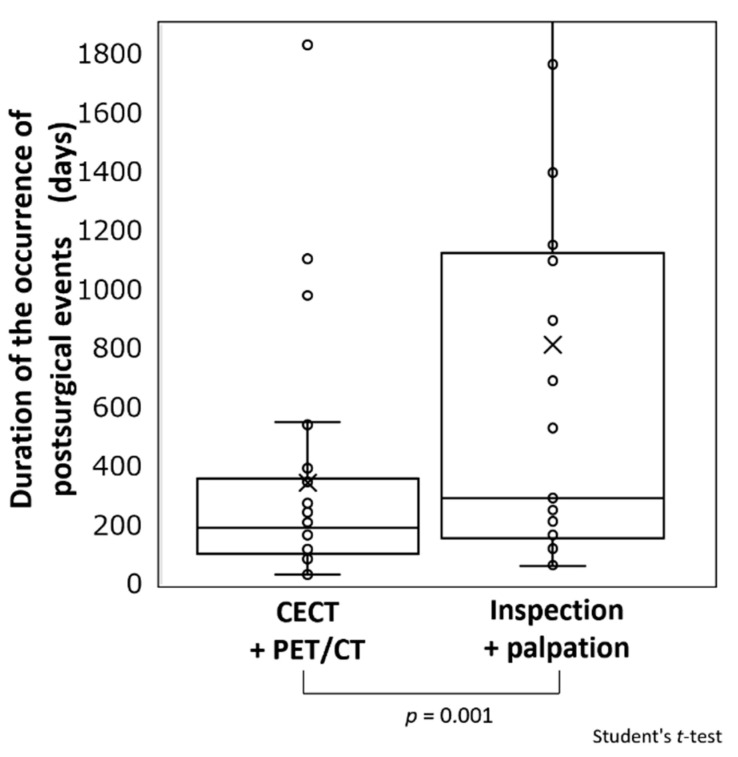
Comparisons of periods until occurrence of postsurgical events based on detection method. The mean times of occurrence of postsurgical events were 345.1 days (Q_2/4_ 190.5 days, Q_3/4_ 359.3 days) for those detected by FDG-PET/CT and CT and 839.3 days (Q_2/4_ 294 days, Q_3/4_ 1139.5 days) for those detected by visual examination and palpation. Thus, events detected visually and by palpation had significantly later onset compared with those detected by imaging (*p* = 0.001). In the figure, “×“ indicates the mean value and “○“ indicates the event occurrence per patient.

**Figure 6 cancers-13-05843-f006:**
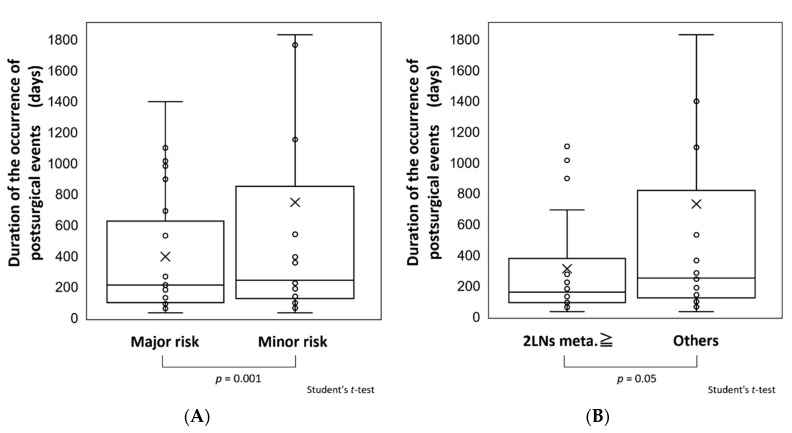
(**A**). Comparisons of periods until occurrence of postsurgical events in the major and minor risk groups. The mean times of occurrence of events were 396.4 days (Q_2/4_ 213 days, Q_3/4_ 625 days) and 747 days (Q_2/4_ 224 days, Q_3/4_ 851.5) in the major and minor risk groups, respectively, with significantly earlier occurrence in the major risk group (*p* = 0.001). (**B**). Comparisons of periods until occurrence of postsurgical events in patients with metastasis to <2 or ≥2 cervical lymph nodes (LNs). The mean times of occurrence of events were 312.2 days (Q_2/4_ 160 days, Q_3/4_ 379.3 days) and 730.9 days (Ｑ_2/4_ 252 days, Q_3/4_ 820 days) in patients with metastasis to <2 or ≥2 LNs, respectively, with significantly earlier occurrence in patients with ≥2 metastases (*p* = 0.05). In the figure, “×” indicates the mean value and “○” indicates the event occurrence per patient.

**Table 1 cancers-13-05843-t001:** Characteristics of patients with OSCC who were treated surgically (*n* = 324).

Variable	*n*	%
Sex, male, *n* (%)	190	(58.6)
Age, mean (SD) y	66.6	(13.8)
Age, median y	69	
Age group, *n* (%)		
<65 y	125	(38.6)
≥65 to <75 y	96	(29.6)
≥75 y	103	(31.8)
Primary site, *n* (%)		
Tongue	156	(48.1)
Lower gingiva	69	(21.3)
Upper gingiva	44	(13.6)
Buccal mucosa	24	(7.4)
Oral floor	20	(6.2)
Lip	6	(1.9)
Palate	5	(1.5)
Clinical T stage, *n* (%)		
Tis	13	(4.0)
T1	52	(16.0)
T2	86	(26.5)
T3	56	(17.3)
T4a	114	(35.2)
T4b	3	(0.9)
Clinical N stage, *n* (%)		
N0	190	(58.6)
N1	65	(20.1
N2b	54	(16.7)
N2c	12	(3.7)
N3b	3	(0.9)
Clinical Stage, *n* (%)		
Stage 0	13	(4.0)
Stage 1	51	(15.7)
Stage 2	60	(18.5)
Stage 3	66	(20.4)
Stage 4a	127	(39.2)
Stage 4b	7	(2.2)
Major risk group, *n* (%)	97	(29.9)
Major risk factor, *n* (%) overlapping distribution		
Extranodal extension	26	(8.0)
Positive or close margins	12	(3.7)
Metastasis of 2 ≥ LNs	41	(12.7)
YK-4C or YK-4D	32	(9.9)
Treatment, *n* (%)		
Surgery only	254	(78.4)
Surgery + postsurgical treatment	70	(21.6)
Postsurgical chemotherapy with cisplatin	12	
Postsurgical chemotherapy with cetuximab + paclitaxel	15	
Postsurgical radiation	10	
Postsurgical chemoradiation with cisplatin	22	
Postsurgical chemoradiation with cetuximab	7	
Postsurgical chemoradiation with cetuximab, followed by chemotherapy with cetuximab + paclitaxel	4	
Adjuvant therapy (S-1 or UFT)	129	(39.8)
Postsurgical event, *n* (%)	65	(20.1)
Death in 5-year period, *n* (%)	55	(17.0)

**Table 2 cancers-13-05843-t002:** Mortality of patients with OSCC with major risk factors after surgical treatment (*n* = 324).

Item	Univariate Analysis			Multivariate Analysis		
	Crude HR	95% CI	*p* Value ^a^	Adjusted HR	95% CI	*p* Value ^a^
YK-4C and 4D	1.068	0.426–2.681	0.888	1.276	0.504–3.230	0.607
Positive or close margin	1.161	0.283–4.764	0.836	1.401	0.339–5.786	0.641
ENE	3.285	1.696–6.366	<0.001	1.676	0.769–3.650	0.194
Metastasis to ≥2 LNs	4.006	2.278–7.043	<0.001	3.342	1.720–6.492	<0.001

HR: hazard ratio, CI: confidence interval; ^a^ Cox-proportional hazard model.

**Table 3 cancers-13-05843-t003:** Postsurgical events of patients with OSCC with major risk factors after surgical treatment (*n* = 324).

Item	Univariate Analysis			Multivariate Analysis		
	Crude HR	95% CI	*p* Value ^a^	Adjusted HR	95% CI	*p* Value ^a^
YK-4C and 4D	0.853	0.426–2.528	0.908	1.136	0.403–3.201	0.810
Positive or close margin	1.992	0.619–6.411	0.248	2.100	0.647–6.817	0.217
ENE	4.397	2.287–8.457	<0.001	2.425	1.101–5.339	0.028
Metastasis to ≥2 LNs	4.175	2.289–7.613	<0.001	2.955	1.437–6.078	0.003

HR: hazard ratio, CI: confidence interval; ^a^ Cox-proportional hazard model.

## Data Availability

The datasets generated and/or analyzed during the study are available from the corresponding author upon reasonable request.

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
