# Peer review of "Surveillance for Patients with Oral Squamous Cell Carcinoma after Complete Surgical Resection as Primary Treatment: A Single-Center Retrospective Cohort Study"

_cancers, 2021, doi:10.3390/cancers13225843_

Round 1

Reviewer 1 Report

"We have used these risk factors since 2014 and we have found that patients with these factors have significantly poorer outcomes and that these outcome are significantly improved by combination therapy with cetuximab and paclitaxel [4, 5]."

--> This approach is not evidence based and not covered by common guidelines

"To clarify the appropriate period, timing, methods for surveillance of patients with OSCC based on the individual risk for recurrence and/or metastasis, we performed a retrospective cohort study after complete surgical resection of OSCC as primary treatment. "

--> the goal to clarify this big issue is not possible by conducting a retrospective study. 

The paper adresses an important issue of HNSCC oncology. But the article presents with some major flaws:

  1. a retrospective study may not be suitable to clarify the best surveillance strategy
  2. the discussion is missing an explanation why FDG positive recurrences were not detected by examination and palpation.
  3. deep body may not be suitable for the oral cavity or neck lymph nodes, as these antomic regions are good visualized by direct endoscopic view and repeated ultrasound examinations.
  4. PET -CT can be benificial to detect vital tumour cells after primary chemoradiation or to detect primary tumors in initially cancer of unknown primary. 
  5. The ratio betwenn advanced tumor stages (T4a and N+) and low rate of postsurgical radiation is not clearly explained to the reader and seems to be out of regular treatment recommendation. 
  6. the study interval of 14 years may be too long to provide comparable imaging quality especially when it comes to CT and PET/CT. 

To sum up, I would suggest to adress some of the issues mentioned above and to resubmit to another journal as the imact of the present paper is not suitable for CANCERS in my opinion. The article may benefit from professional english review. As HNSCC surgeon I don´t see the necessity to perform 3 PETs within 1 year as PET provides a lot of false positive results especially in the oral cavity due to motion artefacts of the tongue. The larynx after chemoradiation is a good target for PET and others too. 

The authors mention the NCCN guideline several times but do not comment on other national guidelines. Maybe a review of surveillance guidelines of oral cancer may be helpful to provide a broader overview of existing strategies. But if the authors like to change the NCCN or to provide alternative surveillance strategy the should think about a prospective study. 

Author Response

Response to reviewer 1

Q: "We have used these risk factors since 2014 and we have found that patients with these factors have significantly poorer outcomes and that these outcome are significantly improved by combination therapy with cetuximab and paclitaxel [4, 5]."

--> This approach is not evidence based and not covered by common guidelines

Response: Postsurgical local and cervical irradiation and/or systemic chemotherapy are considered to be used for invisible tumor cells in high-risk patients. Such treatments are recommended by several guidelines. We recognize that surgical resection of the primary tumor and cervical lymph node dissection is only “debulking” of the visible tumor, but additional local and systemic treatment should be performed in high-risk patients as a preemptive treatment for locoregional and distant micrometastases. Our concept might be the same as that of the guidelines for the treatment of patients with locoregional and distant micrometastases although the tumor cells are not visible clinically or on imaging at the treatment.

Based on our experience and the data in treating many patients with oral cancer, oral cancer patients with the risk factors described here are considered to have invisible locoregional and/or distant micrometastases at the time of surgery. It is clear that the effector (or therapeutic agent)/target ratio is directly related to the therapeutic efficacy of both cytotoxic anticancer agents and molecularly targeted agents as immunotherapy. Therefore, we believe that preemptive therapy is necessary while the tumor cells disseminated throughout the body are still microscopic. As described in detail in Ref. [4, 5], this treatment is not a postsurgical preventive therapy, but a preemptive therapy for metastatic tumors. Our previous article Ref. [4, 5] clearly showed the evidence of the statistical effect of our approach. We slightly corrected the description. (in introduction section, line 61, page 2)

Q: "To clarify the appropriate period, timing, methods for surveillance of patients with OSCC based on the individual risk for recurrence and/or metastasis, we performed a retrospective cohort study after complete surgical resection of OSCC as primary treatment. "

--> the goal to clarify this big issue is not possible by conducting a retrospective study. 

Response: We believe that the goal of the study should be clearly shown. However, we also do not think that the results of this study alone can reach the goal we proposed. For establishing the surveillance methods for postsurgical patients with oral cancer, the accumulation of high-quality retrospective observational cohort research studies is important. In making clinical practice guideliness, it is proposed that the importance of high-quality retrospective and prospective observational studies for CQs for which randomized interventional studies are ethically difficult. In our opinion, the interventional prospective trials, which are usually used for the development of treatment methods, are not appropriate for establishing the surveillance methods. If postsurgical contrast-enhanced CT, MRI, and PET/CT are hypothesized to be ineffective, and only history, visual examination, palpation, and ultrasonography are considered to be standard surveillance methods, the usefulness of these imaging could be investigated in a randomized intervention. However, such a study might be ethically difficult. In fact, the recommendations for postoperative surveillance methods cited in the NCCN guideliness are also based on observational studies. We slightly corrected the description. (in discussion section, line 353, page 12)

Q: The paper adresses an important issue of HNSCC oncology. But the article presents with some major flaws:

  1. a retrospective study may not be suitable to clarify the best surveillance strategy

Response: As mentioned above, we believe that high-quality retrospective observational studies are also important for establishing the better surveillance strategy.

  1. the discussion is missing an explanation why FDG positive recurrences were not detected by examination and palpation.

Response: Postsurgical events that can be detected by palpation and visual examination are limited to local recurrences confined to areas exposed to the oral cavity and lymph node lesions that can be palpable from the external surface. It is difficult to detect metastatic lymph nodes behind the sternocleidomastoid muscle or the upper area around the internal jugular vein (level II area) by palpation or visual examination even in the neck without dissection. We added the explanation. (in discussion section, line 332, page 11)

  1. deep body may not be suitable for the oral cavity or neck lymph nodes, as these antomic regions are good visualized by direct endoscopic view and repeated ultrasound examinations.

Response: Yes, we understand the usefulness of endoscopy and ultrasonography, and we routinely use them for surveillance. However, we can observe only surface and sub-surface areas by these devices. As for the local and cervical areas in the patients who are reconstructed by the vascularized skin and muscular flap, recurrences at the deep area cannot be detecable by only visual and palpation, but also endoscopy and ultrasonography. We slightly corrected the description. (in discussion section, line 336, page 11)

  1. PET -CT can be beneficial to detect vital tumour cells after primary chemoradiation or to detect primary tumors in initially cancer of unknown primary. 

Response: Yes, we and everyone understand that PET -CT is beneficial to detect the vital tumor cells that the reviewer points out. Here, we demonstrated that PET/CT was also useful for detecting metastatic disease in deep tissues and distant organs, even in patients after primary surgery. We slightly corrected the description. (in discussion section, line 340, page 11)

The ratio between advanced tumor stages (T4a and N+) and low rate of postsurgical radiation is not clearly explained to the reader and seems to be out of regular treatment recommendation. 

Response: As the reviewer points out, we understand that the NCCN guidelines stated that "postoperative treatment should be considered" for T4a and N+. As shown in Table 1, T4a is the most common T stage category in our institution, accounting for 35% of cases. However, we think that local tumor size and progression are not prognostic factors if surgical complete resection is performed with adequate surgical margins. Therefore, we do not include locally advanced cancer in the major risk category. Furthermore, we think that the biological aggressiveness and prognosis between >N2 patients and N1 patients are quite different. Therefore, we do not include N1 patients in the major risk category. The NCCN guidelines dose not show that postoperative treatment (postsurgical radiation) is recommended for T4a and N+, but postoperative treatment should be considered for T4a and N+. Therefore, we do not think that our treatment is not out of regular treatment recommendation from the guidelines. We slightly modified the description. (in introduction section, line 62, page 2)

  1. the study interval of 14 years may be too long to provide comparable imaging quality especially when it comes to CT and PET/CT. 

Response: We have our own PET center in our institute from 2005 and also have our own cyclotron to produce 18F-FDG, and provide high quality images. Although the quality of images has improved recently, we believe that the images and diagnosis of 14 years ago have high quality enough for comparison.

Q: To sum up, I would suggest to adress some of the issues mentioned above and to resubmit to another journal as the imact of the present paper is not suitable for CANCERS in my opinion.

Response: Thank you very much for your critical reading and comments to improve our manuscript.

Q: The article may benefit from professional english review.

Response: This manuscript is edited by an English native speaker with medical and scientific knowledge before submission. This is our best. If you find some specific grammatical errors or inappropriate descriptions, please point out individually.

Q: As HNSCC surgeon I don’t see the necessity to perform 3 PETs within 1 year as PET provides a lot of false positive results especially in the oral cavity due to motion artefacts of the tongue. The larynx after chemoradiation is a good target for PET and others too. 

Response: We do not mean the necessity for 3 times PET/CT per year in the conclusion of this paper. There might be confusion because we described "FDG-PET/CT or CECT at 3 and 6 months and 1 year after surgery" in manuscript. Our intention is to perform PET/CT at 3-6 months postoperatively and once a year thereafter, and to consider CECT as an option in between PET/CT. We modified the description to make it easier to understand (in simple summary, abstract section, page 1 and conclusion section, line 372, page 12). We understand the characteristics of PET as the reviewer mentioned, and utilized PET to make better diagnosis accordingly.

Q: The authors mention the NCCN guidelines several times but do not comment on other national guidelines. Maybe a review of surveillance guidelines of oral cancer may be helpful to provide a broader overview of existing strategies.

Response: We believe that the NCCN guidelines is the most internationally recognized and utilized guidelines. Therefore, we are using it as the standard for our clinical diagnosis and treatment, and the discussion in this manuscript. We have also been involved in publishing the guidelines for oral cancer in Japan, and of course we utilize the guidelines in diagnosis and treatment. The guidelines in Japan, the NCCN guidelines, and the ESMO guidelines do not differ greatly. Guidelines should be used as a benchmark, and diagnosis and treatment should be generally performed in accordance with them particularly in surgical treatment. In scientific thinking, if there are multiple standards, considerations would become diffuse rather than broad. Then, we proceeded the discussion using a single standard, the NCCN guidelines. We slightly modified the description. (in discussion section, line 316, page 11)

Q: But if the authors like to change the NCCN or to provide alternative surveillance strategy the should think about a prospective study. 

Response: We think that there are no clinical studies in which the specific aim of the study is to change the guidelines. The guidelines should be changed based on the data accumulated from observational and interventional studies. As we mention above, prospective interventional studies are not suitable for establishing the protocol for postoperative surveillance, and we believe that accumulation of the high-quality observational studies is important. We slightly modified the description. (in discussion section, line 353, page 12)

Reviewer 2 Report

Thanking in consideration the literature data there are such paper talking about the overall survival after complete surgical excision of OSCC. In particular the main interesting  thing  of this paper, by my point of view, is the accurate presentation of the method explained by table and figure. I think that it is difficult find out paper of a a single center retrospective cohort study with this number of patients underwent to primary surgery excision.  The analysis of the percentage of local recurrence such as outcomes and postsurgical events was accurate investigated by performing good Statistical analysis.   The wide use of FDG-PET/CT is so debated and there are different experiences published not with such accurate data.   Therefore, in conclusion, I believe that the work has the right scientific impact, especially because it analyzes surgically treated cases, taking in consideration all the complication during follow up and especially analyzing the importance of FDG-PET/CT, that can help clinician to intercept recurrence or metastasis of the disease .  

Author Response

Response to reviewer 2

Thanking in consideration the literature data there are such paper talking about the overall survival after complete surgical excision of OSCC. In particular the main interesting thing of this paper, by my point of view, is the accurate presentation of the method explained by table and figure. I think that it is difficult find out paper of a a single center retrospective cohort study with this number of patients underwent to primary surgery excision. The analysis of the percentage of local recurrence such as outcomes and postsurgical events was accurate investigated by performing good Statistical analysis. The wide use of FDG-PET/CT is so debated and there are different experiences published not with such accurate data. Therefore, in conclusion, I believe that the work has the right scientific impact, especially because it analyzes surgically treated cases, taking in consideration all the complication during follow up and especially analyzing the importance of FDG-PET/CT, that can help clinician to intercept recurrence or metastasis of the disease .

Response:We thank you for this pertinent comment.

Reviewer 3 Report

Fukumoto et al. retrospectively analyzed the importance of surveillance for patients who had undergone complete surgical resection and sought better postoperative management strategies, focusing on postoperative events, such as local recurrence, metastases, or secondary cancers.

They found that half of the postsurgical events took place within 200 days, and 75% of them within 400 days. They recommended imaging studies in combination with physical examination for earlier detection of the postsurgical events based on detection modalities.

Major issue

Although among major risk factors, extranodal extension and multiple LN metastases stratified mortality rates and postsurgical events, the histological grading (differentiation status) did not correlate with poorer outcomes. 

The reviewer felt that this was somewhat strange or due to other confounding factors. Postsurgical adjuvant therapies appear to vary significantly depending on patients or the treatment date in the authors’ institution (of course, it is inevitable). The authors started using the YK-stratification strategy, but the study group encompassed OSCC patients treated from 2007 to 2020. Poor differentiation capacity principally confers poorer outcomes due to locally aggressive biological behaviors. The data may evoke confusion in the readers. They should have been excluded from the retrospective analysis, or the authors have to explain explicitly for scientific integrity or to make sense.

Author Response

Response to reviewer 3

Q: Although among major risk factors, extranodal extension and multiple LN metastases stratified mortality rates and postsurgical events, the histological grading (differentiation status) did not correlate with poorer outcomes. The reviewer felt that this was somewhat strange or due to other confounding factors.

Response: We applied the mode of invasion (YK classification) rather than the degree of differentiation as a histological assessment as a postsurgical risk assessment. As the reviewer pointed out, the mode of invasion was not a risk factor for patient prognosis in this study. In all patients with high YK classification (YK4C and 4D), surgical treatment (primary resection and neck dissection) might be appropriately performed. Therefore, OS in all patients was good. In addition, the application of Cmab-based therapy as a preemptive treatment for high-risk group improved the prognosis, resulting the selection bias for therapy. We added a description to explain this fact. (in results section, line 199, page 7)

On the other hand, multiple LN metastases were significantly associated with poor prognosis in multivariate analysis, despite postsurgical treatment was performed in many patients. In such patients, even after complete surgical resection at the primary treatment and subsequent postsurgical treatment, locoregional recurrence in the areas that were difficult to salvage, and/or distant metastasis occurred. A number of invisible tumor cells might exist at the time of multiple LN metastases.

Q: Postsurgical adjuvant therapies appear to vary significantly depending on patients or the treatment date in the authors’ institution (of course, it is inevitable).

Response: As the reviewer pointed out, the study period was long (14 years), therefore the methods for postsurgical treatment varied. We employed the methods recommended in the newest guidelines and the available drugs on each period.

Q: The authors started using the YK-stratification strategy, but the study group encompassed OSCC patients treated from 2007 to 2020. Poor differentiation capacity principally confers poorer outcomes due to locally aggressive biological behaviors. The data may evoke confusion in the readers. They should have been excluded from the retrospective analysis, or the authors have to explain explicitly for scientific integrity or to make sense.

Response: As the reviewer pointed out, the mode of invasion was not extracted as a risk factor for mortality and the occurrence of the events in this study. As we mentioned above, we added a description to explain this fact. (in results section, line 199, page 7)

Round 2

Reviewer 1 Report

Dear Authors, thank you for providing this revision with clarification of all adressed issues. As I mentioned surveillance studies are a very important HNSCC topic and finally the study in the present form shows a very professional way to consider risc factors for surveillance strategy.

I still have doubt, if the preemptive treatment with cetuximab is reasonable but this does not influence too much the results and conclusions of this paper. 

Reviewer 3 Report

The authors addressed the concerns appropriately.